# Understanding South Korea’s Response to the COVID-19 Outbreak: A Real-Time Analysis

**DOI:** 10.3390/ijerph17249571

**Published:** 2020-12-21

**Authors:** Eunsun Jeong, Munire Hagose, Hyungul Jung, Moran Ki, Antoine Flahault

**Affiliations:** 1Institute of Global Health, University of Geneva, 1211 Geneva, Switzerland; 2Department of Cancer Control and Policy (DCCP), Graduate School of Cancer Science and Policy, National Cancer Centre (NCC), Goyang 10408, Korea; moranki@ncc.re.kr; 3Institute of Global Health, Faculty of Medicine, University of Geneva, 1205 Geneva, Switzerland; Antoine.Flahault@unige.ch

**Keywords:** South Korea, coronavirus, COVID-19, global health, case study, epidemiology, non-pharmaceutical interventions, health systems

## Abstract

This case study focuses on the epidemiological situation of the COVID-19 outbreak, its impacts and the measures South Korea undertook during the first wave of the COVID-19 pandemic. Since the first case was confirmed on 20 January 2020, South Korea has been actively experiencing the COVID-19 outbreak. In the early stage of the pandemic, South Korea was one of the most-affected countries because of a large outbreak related to meetings of a religious movement, namely the Shincheonji Church of Jesus, in a city called Daegu and North Gyeongsang province. However, South Korea was held as a model for many other countries as it appeared to slow the spread of the outbreak with distinctive approaches and interventions. First of all, with drastic and early intervention strategies it conducted massive tracing and testing in a combination of case isolation. These measures were underpinned by transparent risk communication, civil society mobilization, improvement of accessibility and affordability of the treatment and test, the consistent public message on the potential benefit of wearing a mask, and innovation. Innovative measures include the mobile case-tracing application, mobile self-quarantine safety protection application, mobile self-diagnosis application, and drive-thru screening centres. Meanwhile, the epidemic has brought enormous impacts on society economically and socially. Given its relationship with China, where the outbreak originated, the economic impact in South Korea was predicted to be intense and it was already observed since February due to a decline in exports. The pandemic and measures undertaken by the government also have resulted in social conflicts and debates, human-right concerns, and political tension. Moreover, it was believed that the outbreak of COVID-19 and the governmental responses towards it has brought a huge impact on the general election in April. Despite of the large outbreak in late February, the Korean government has flattened the COVID-19 curve successfully and the downward trend in the number of new cases remained continuously as of 30 April. The most distinctive feature of South Korea’s responses is that South Korea conducted proactive case finding, contacts tracing, and isolations of cases instead of taking traditional measures of the containment of the epidemic such as boarder closures and lockdowns.

## 1. Introduction

A novel coronavirus, namely COVID-19, caused by severe acute respiratory syndrome coronavirus 2 (SARS-Cov-2) has emerged in Wuhan, China in December 2019. A few weeks later, the World Health Organization (WHO) announced the outbreak of a new virus. As the epidemic has spread across the world at an unprecedented rate, the WHO declared, on the 31st of January 2020, the 2019 novel coronavirus as a Public Health Emergency of International Concern [1]. Neighbouring countries to China such as Thailand and South Korea were the first countries to report cases before the virus started to spread worldwide. South Korea declared its first case on the 20th of January 2020 [2]. As the number of cases has rapidly soared due to the large outbreak related to religious meetings of the Shincheonji Church of Jesus [3], it became the second-most affected country in the world after China in late February. However, as the Republic of Korea combated the epidemic actively by taking proactive measures to reduce the number of daily new cases, the country handled the spread of COVID-19 impressively as soon as the first case was declared in its territory. As of April 23, it ranked 29th among the countries most affected by the virus, with 10,702 declared cases [4].

In 2015, South Korea was affected by the Middle East Respiratory Syndrome Coronavirus (MERS-CoV). It recorded 186 cases, including 38 fatalities. The 2015 MERS outbreak revealed the weakness of South Korea’s healthcare system to tackle emerging and re-emerging infectious diseases. Since then, the Republic of Korea has made a change in the systems and policies to be capable of tackling the epidemics successfully [5]. As soon as the COVID-19 outbreak was announced in South Korea, a series of policies and interventions to contain the dissemination of the coronavirus disease were adopted, promptly and effectively. Research to develop a test kit was launched in mid-January, right after the Chinese government shared the genetic sequences of the virus [6]. Thus, when cases were rising up due to community transmission, it has already been able to detect and trace infected people and isolate them swiftly. South Korea remarkably controlled and flattened its curve without any national lockdown, even in Daegu and North Gyeongsang Province where most cases occurred [7]. All these reasons put South Korea as an interesting, but also important, country to analyse and discuss in order to have a clearer comprehension of the measures undertaken as there is no harmonized and coordinated measures worldwide. Scrutinizing how each country has responded to COVID-19 and its consequences may broaden our insights into the COVID-19 pandemic. The present case study will, first, identify the evolution of the outbreak in South Korea. Thereafter, non-pharmaceutical intervention measures undertaken, economic, political, and social impacts, and mathematical prediction will be discussed.

## 2. Methodology

We conducted the case study by analysing the early responses of South Korea to the COVID-19 pandemic. It is a real-time analysis of the situation regarding the Covid-19 epidemic in South Korea as it was conducted during the ongoing pandemic. At the initial phase of the pandemic, the sources of data were limited and there were few peer-reviewed scientific researches available. Therefore, we utilized data from governmental websites such as Korea Centres for Disease Control and Prevention and Korea Ministry of Health and Welfare, governmental reports, WHO publications, scientific articles, and conventional media. The study highlights and analyses responses of the South Korean government through the scientific knowledge and resources we had in April 2020. Based on data provided by the Korean government, Korea Centres for Disease Control and Prevention, the United States Centres for Disease Control and Prevention, and the COVID tracking project, we were able to draw various figures by Microsoft Excel. In addition, the schematic diagram was developed to illustrate the non-pharmaceutical intervention measures. As this case study focuses on the first of the COVID-19 pandemic, all the epidemiological data presented are dated between January and April 2020.

## 3. Findings

### 3.1. Case Presentation

#### 3.1.1. General Description

South Korea is an East Asian country with 51 million inhabitants and half of the population is concentrated in the capital Seoul and its metropolitan area. The density of the population is estimated to be around 503 people per square kilometres, while the density of Seoul is approximately 17,000 people per square kilometres. The median age of the population is 42 years and the proportion of people older than 65 years is 15.5%. The life expectancy of the population is around 83 years. The country has a land border with North Korea and is surrounded by the Yellow Sea and the East Sea that are situated between South Korea and China and South Korea and Japan respectively [8]. According to the Organization for Economic Cooperation and Development (OECD), South Korea is a wealthy and developed country with access to high technologies. The country is considered as the 11th largest economy in the world [9]. Regarding the country’s economic system, the Republic of Korea relies mainly on a strategy of exporting goods. The top export partner is China, its neighbouring country [8]. The climate of South Korea is temperate with four distinct seasons. The annual mean temperature ranges from 10 °C to 16 °C. The coldest month is January, with a mean temperature ranges from −6 °C to 7 °C, while the warmest month is August, with a mean temperature range from 23 °C to 27 °C. The outbreak of the novel coronavirus in South Korea happened in winter, which is a cold and dry season [10]. Concerning the political aspect, South Korea is a democracy with a president, Moon Jae-In since May 2017. Moon Jae-In is from the Democratic Party of Korea, which is known as a centre-left party. The country has a unicameral parliament composed of 300 members elected for four years. Currently, the Democratic Party of Korea, the president’s party, is the most represented in the National Assembly [11].

#### 3.1.2. Healthcare System

Overall, South Korea’s health care system is described as being one of the greatest. The government expenditure for the health system was about 7.6% of its GDP in 2017 [12]. According to Bloomberg’s official ranking, South Korea has the fifth most efficient health care system in the world [13]. In addition, South Korea is the fifth-highest country with Intensive Care Unit (ICU) beds per capita and it has 10.6 beds per 100,000 inhabitants [14]. It was reported that it had 12.6 hospital beds per 1000 inhabitants in 2018, ranked second among OECD countries [9]. As the Republic of Korea is a member state of the International Health Regulations (IHR), in 2017, the joint external evaluation mission took place in order to assess its preparedness for a public health emergency. The mission concluded that “the Republic of Korea has highly sophisticated systems and capacities in place to address emerging and re-emerging infectious disease threats and public health emergencies” [15] (p. 1).

South Korea achieved universal health coverage for its population in 1989. In 2000, the National Health Insurance (NHI) was introduced as the only insurance of the country with a uniform contribution schedule and benefits coverage for the citizens [16]. However, one of the major issues of the public health system is that “health-care delivery relies heavily on private providers” [16]. As a result of providers’ behaviour seeking profit, there is an increase in demand for new services and technologies that are not included in the NHI benefit package and it is one of the main reasons for the high level of Out-of-Pocket payments [16]. It was also pointed out the public health sector is poor both in terms of quantity and quality. There are insufficient public health facilities and workforce, and a shortage of finance [17]. Moreover, an increase in the cost of healthcare and overuse of medical services have been major problems caused by not only aging and a rise in the number of patients suffering from chronic diseases but also the inefficient healthcare system [18]. Through the COVID-19 outbreak, South Korea has reconfirmed a perennial problem, the lack of health workforce in general. The army doctors and nurses, volunteers, and public health doctors have been dispatched to affected areas to alleviate a shortage of medical personnel. Especially during the large outbreak in Daegu, 750 public health doctors were newly recruited and sent to serve in Daegu [19]. Public health doctors are male doctors who work in remote areas for three years instead of military service under a substitute military service system and they have been playing a significant role in the containment of the outbreak [20]. Meanwhile, South Korea’s government or the National Health Insurance cover all costs that arose from a diagnostic test to hospital admission for its population and foreigners, if it’s related to the coronavirus disease [21].

#### 3.1.3. Epidemiological Situation of the Country Regarding COVID-19

The Republic of Korea declared its first confirmed case of COVID-19—a Chinese visitor who came from Wuhan—on 20 January 2020 [2]. Thereafter, the virus spread very slowly in the country with only a small number of new cases during the first few weeks of the outbreak. In mid-February, South Korea counted 28 cases in its territory, however, as of the 19 February 2020, the epidemic began to accelerate with a higher number of new cases every day. The main health authority for the COVID-19, the Korea Centres for Disease Control and Prevention (KCDC), decided to raise the level of infectious disease alert to “red”, which is the highest level, on 23 February [22]. This action introduced more strengthened public health measures, for example, social distancing, mask-wearing campaigns and mass diagnostic tests to contain the virus and minimize local propagation in advance. However, the country quickly encountered huge unexpected outbreaks in the local areas. On the 29 February 2020, the country reached its peak of new cases in a single day, reporting 909 cases [23]. The dramatic increase in the number of cases was mainly derived from a large outbreak in Daegu, the fourth biggest city with 2.5 million people, and North Gyeongsang Province [24]. It was turned out that this large outbreak was associated with a fringe religious sect called Shincheonji Church of Jesus. The epidemiological investigation revealed that a massive propagation of the virus took place among worshipers during services [24]. However, the rapid increase in new cases had turned downward after the number of new cases peaked at 909 cases. As the number of new cases has rapidly dropped, the curve of confirmed cases has flattened [25]. Figure 1 illustrates the trend in new and cumulative numbers of COVID-19 cases and deaths from late January to April [26,27]. As of 30 April 2020, South Korea has reported 10,765 confirmed cases of COVID-19 with 247 deaths which occurred mostly among those over 60 years old (92%). 63.9% of the cases were found in Daegu, and the Shincheonji-related cluster outbreak accounted for 48.7% of total cases across the country [28]. In spite of spikes in the daily new cases from late February to early March, the number of deaths remained stable as seen in Figure 1. In addition, the number of new cases has dropped to around 10 per day and as of 30 April it was reported that 9,059 patients have fully recovered that is 84.2% of the total cases [25].

619,881 diagnostic tests have been conducted across the country (Figure 2) until at the end of April [26,27]. However, it appeared that not only the total numbers of tests but also the early implementation matter. The importance of early reactive case detection becomes clear when we compare cumulative numbers of tests in South Korea and the United States as seen in Figure 3 [26,29,30]. Both countries have confirmed the first COVID-19 case on 20 and 21 January respectively [2,29]. While South Korea expanded case detection immediately, the United States increased the number of tests at late February where the number of confirmed cases was soaring exponentially [29]. Figure 4 shows sex- and age-disaggregated data on reported COVID-19 cases as of 30 April. The male to female ratio was confirmed approximately 4:6 (M:F) and young adults aged 20–29 was the age group with the highest rate of infection, which is 27.42% of total infections [25].

### 3.2. Management and Outcome

#### 3.2.1. Non-Pharmaceutical Intervention Measures

The non-pharmaceutical interventions (NPIs) are public health measures aiming at decreasing transmission by lowering contact rates [31]. As there is no vaccine available against COVID-19, South Korea has been implementing proactive and distinctive non-pharmaceutical interventions such as massive case finding and tracing, meticulous managing exposed or confirmed cases, and providing the public a consistent message to wear a mask and its potential benefit. The massive case finding and tracing could be possible thanks to its ability to produce test kits domestically when community transmission has started [32] and drive-through screening centres that boosted the country’s capacity in testing [33]. It was known that South Korea prepared itself to be capable of producing test kits in collaboration with the private sector at the early stage of the outbreak [32]. Moreover, the massive case finding and tracing have led to rigorous interventions to manage exposed and confirmed cases [34]. For instance, the government introduced the mobile application to monitor self-isolated patients and the Living treatment centres to isolate mild or asymptomatic patients. The rapid and innovative responses of the government were believed to be due to its experience of MERS in 2015 [32]. The experience of MERS has left lessons learnt and affected the way it is tackling the epidemic of COVID-19. The government was condemned for lack of transparent information to the public, which significantly contributed to the spread of the disease. It was claimed that many cases would have been saved by notifying contacts that they were exposed to the confirmed case and providing information on travel history of confirmed cases so that the public could avoid visiting affected places. Consequently, laws passed to allow authorities to trace infected individuals and disclose information on the cases to the public. Therefore, when the COVID-19 epidemic started, the country was able to set up the system for case tracing quickly [35]. Although it raised the concerns of human rights, the public information disclosed underpinned transparent risk communication during the COVID-19 outbreak. In addition, South Korea has been containing the outbreak remarkably without lockdowns or border closures while respecting the freedom of movement of populations and reducing the economic impact. To illustrate various measures and strategies other than well-known massive testing, we have developed the schematic diagram (Figure 5) and the main features are further described in detail in below.

As it is illustrated in Figure 5, South Korea took several measures for rigorous case isolations. Confirmed cases were hospitalized in the health facilities or the Living Treatment Centres based on the severity of illness [34]. The Living Treatment Centres, quarantine facilities, were introduced on 2 March 2020, to isolate confirmed cases not requiring hospitalization to minimize the community transmission while reducing the burden on the healthcare system [37]. To monitor patients efficiently in the centres without unnecessary contacts, medical personnel who were assigned to the Living Treatment Centres were using the Self-Quarantine Safety Protection Application through which patients input individual symptoms twice a day. It was also mandatory for those who were under self-quarantine to download this app or the Self-Diagnosis App [38].

Along with the case isolation and treatment, an epidemiologic survey was conducted in each case. The travel history of patients was traced thoroughly using data such as credit card usage, CCTV, and mobile GPS to conduct environmental disinfection and identify contacts [36]. The public information disclosed containing cases’ travel history, in turn, was utilized by companies or individuals to develop the mobile contact tracing apps [39]. The contacts identified had to be self-isolated under monitoring by local governments through the mobile application or phone. During self-quarantine, if a symptom was developed newly, it was directly notified to a public officer through this application. The mobile application was used not only to monitor symptoms but also to spot locations to know whether patients comply with the rule [38]. If those under self-quarantine were found at any place other than their home or a quarantine facility, they would face a fine or imprisonment [36].

The early detection and isolation of cases were underpinned by civil society mobilization, improvement in accessibility and affordability to the screening test and treatment, and prevention of the spread of the outbreak in communities and healthcare facilities. The government has mobilized civil society through risk communication emphasizing the importance of its role and advised people to apply public health measures such as hand washing and wearing a mask and keeping social distance while avoiding large gatherings. Especially, the Ministry of Food and Drug Safety has issued the guideline on the use of masks for the public. It recommended to wear a certified medical mask against COVID-19 from the early phase of the pandemic [40]. Moreover, it has provided the public with transparent information on the outbreak including information regarding confirmed cases through the Regular Briefing of Central Disaster and Safety Countermeasure Headquarters on COVID-19 and press conference, text message alerts, and applications [26,36].

Improvement in accessibility and affordability also played a key role. It was achieved by establishing a great number of screening centres such as drive-through centres [33] and 24 Hours call centres providing consultations and expanding the criteria of the diagnostic test to allow testing asymptomatic people. According to Korea Centres for Disease Control and Prevention, the costs of the treatment and tests of suspected or confirmed cases is covered fully by the National Health Insurance or the government [36].

With a massive case finding and testing, it also monitored the general population by surveillance of pneumonia patients in hospitals and temperature screening at places where people gather such as a train station, a shopping mall, a restaurant, and so on [36]. In order to reduce nosocomial infection of COVID-19, patients with any respiratory symptoms were treated separately in designated hospitals, which can be easily noticeable, so that it can prevent people from exposure to unidentified or confirmed cases. And through the national systems, the International Traveller Information System (ITS) and Drug Utilization Review (DUR), health care facilities were provided with critical information such as patients’ oversea travel history to major countries affected by the outbreak of COVID-19 and whether the patient was a worshiper of Shincheonji Church of Jesus so as to help doctors and nurses to diagnose and take precautions in advance [34,36].

In contrast with the stabilized epidemic in South Korea, European countries and the USA became the epicentres of the pandemic in April where the number of cases and deaths were soaring at terrifying speed [4]. As a result, there was an increase in cases among people arriving from overseas. The Korean government, therefore, decided to conduct the test and put all inbound travellers including Korean citizens arriving from Europe under self-isolation regardless of having symptoms from 1 April [36]. However, there was no travel bans since the emergence of the outbreak except foreigners arriving from Hubei Province, China since 4 February [41]. Also, schools and childcare centres were closed. A new semester was supposed to start on 2 March, but it was postponed a few times due to the persistent possibility of the spread of the disease among students and teachers. As the outbreak and absence of education were prolonged, schools including elementary, middle, and high schools, started opening online schools nationwide since 9 April [42]. Facilities with mass gatherings such as a church, a community child centre, a senior welfare centre were advised to close, however the government has not imposed a national lockdown [36].

#### 3.2.2. Expected or Observed Impact on the Country Economy

This unprecedented situation the world is facing, had a significant impact on the national and international economy. China, where the epidemic of COVID-19 has started, plays a key role in travel and commodity markets and supply chains all over the world. Due to its significant role, the noteworthy economic impact of COVID-19 in China has been seen in other countries before it struck them. Also, as the outbreak was spreading, it was causing economic disruption worldwide [9]. The Korea Development Institute (KDI) revealed that in South Korea the production growth did not decrease in January when the few cases were detected [43]. However, the slowdown of exports appeared and domestic demand weakened in February as the COVID-19 outbreak was spread further. The exports have decreased due to not only a decline in demand from China but also disruption of the supply of immediate goods to produce commodities especially automobiles. In addition to deterioration in external factors, domestic demand decreased as a result of a deterioration in economic sentiment [43].

The OECD indicated that annual global GDP growth is expected to decrease by 0.5% in 2020 with negative growth in the first quarter possibly. However, the decline in global growth could be 1.5% relative to 2.9%, the rate expected before the outbreak, if it spreads worldwide resulting in a severe and longer-lasting outbreak. It was also mentioned that the economic adverse impact will be stronger in South Korea, Japan, and Australia which are highly interdependent [9].

#### 3.2.3. Social and Political Disruption

The COVID-19 outbreak in South Korea has brought a variety of social impacts across the country. South Korea’s government came under political criticism for not blocking all arrivals from China amid the peak of the epidemic in China. As a result, more than 700,000 people have signed a petition for travel bans from all parts of China and the issue became a big political argument [44]. It has not only caused political debates but also affected the result of the election on 15 April 2020. As this particular situation happened just before the parliament’s election which occurs every four years, the management of the epidemic by President Moon Jae-in and the government has influenced the public vote [45]. Unlike criticism on the government earlier, voters applauded President Moon Jae-in and the government for successful responses to the coronavirus outbreak. As a result, the Democratic Party of Korea had a comfortable majority in the parliamentary elections, thanks in part to the management of the health crisis [45].

As the number of the COVID-19 case increased, South Korea was confronted with a lack of protective equipment, such as masks and hand sanitizers, due to insufficient supply, some domestic merchants’ hoarding and panic buying in the early period of the outbreak. Consequently, people have struggled to secure masks [46]. The government took action to control the supply and distribution of masks and imposed penalties on the hoarding of masks. The government started to manage the whole process of production, logistics, and distribution of masks in South Korea and even banned mask exports. For example, ‘the 5days rotation system’ has been implemented for mask distribution, through which people can buy two masks per week from pharmacies on designated days of week relying on their year of birth [47]. To monitor the purchase of a mask and distribute it equally, the protective mask was newly included in the Drug Utilization Review program or DUR, which is a national system to restrain patients from buying the same drug repeatedly. Consequently, “the 5 days rotation system” could be implemented successfully by facilitating the pre-existing system, the DUR [48].

After the massive COVID-19 epidemic was found in Daegu and Gyeongbuk areas, a wide range of thorough investigations were carried out by health authorities and revealed the connection between this outbreak and the Shincheonji religious movements. As a result, all those churches were forcibly closed by the authorities temporarily and even one local government accused them of ‘murder due to wilful negligence’. There was a petition for the Shincheonji church to be dismantled even if freedom of religion could be violated. Inversely, the church’s worshippers alleged that they have been persecuted and stigmatized by society because some of them were dismissed and excluded from their works merely because of their affiliation [49].

Lastly, in spite of its remarkable results, there were growing human-right concerns on intrusive case tracing and disclosure of private information of cases [35]. It was often said that information of cases disclosed might be identifiable. In turn, it could cause a violation of human rights and stigmatization. Also, as enormous and detailed information was provided by the government, people raised concern on consequent psychological effects. The authorities sent unceasing emergency text messages, alerting them to travel history of cases and the importance of personal hygiene and social distance. Therefore, it might cause various psychological effects such as anxiety, tiredness, and insensitivity due to exceedingly frequent alerts [35].

#### 3.2.4. Mathematical Modelling Predictions

Mathematical modelling has significant roles in responding to infectious disease outbreaks and establishing prevention measures. It helps predict the size and duration of the outbreak or the effects of public health interventions even if accessible information is limited. Choi’s work regarding the COVID-19 mathematical modeling was performed right after the Shincheonji outbreak in Daegu, South Korea and it projected the virus’ propagation and the results of interventions during the first wave of the pandemic [50]. Her work illustrated the reproduction number (*R*) of the initial outbreak through the SEIHR model and served as mathematical modeling predictions according to its possible scenarios the reproduction number (*R*) refers to how many secondary individuals can be infected by a primary individual who is thought to be infected [51]. The estimation of reproduction number (*R*) is determined by the probability of transmission and the period of infection transmission. The SEIHR compartment consisting of Susceptible (S), Exposed (E), Infectious (I), Hospital-quarantined (H), and Recovered (R), was used. The model is illustrated in Figure 6.

According to the study, the estimated reproduction numbers (*R*) ranged from 3.539 to 3.476 (based on the confirmed case from 29 February to 4 March 2020). Moreover, if no virus containment measures were introduced in Daegu and North Gyeongsang province (Gyeongbuk), the expected peak point would be 5 April 2020 and the infected number of cases would have reached 22,389 (Figure 7).

Additionally, the study projects that the epidemic would have ended on June 28th, 2020 and total confirmed cases would have reached 4,992,000 [50]. On the other hand, if there were countermeasures to reduce infections, those would have led to decreasing the transmission rate and the infection transmission period alike, since early detection and case isolations are boosted by the measures. Figure 8 illustrates various curves on the base of the timing of containment and the reduced transmission rates according to given scenarios. (Scenario 1: the day of containment measures becomes effective—5 March, the transmission period—4 days, 90% reduced transmission rate, Scenario 2: the day of containment measures becomes effective—March 5th, the transmission period—4 days, 99% reduced transmission rate, Scenario 3: the day of containment measures becomes effective—5 March, the transmission period—2 days, 99% reduced transmission rate, Scenario 4: the day of containment measures becomes effective—29 February, the transmission period—4 days, 90% reduced transmission rate, Scenario 5: the day of containment measures becomes effective—29 February, the transmission period—2 days, 75% reduced transmission rate).

In another study, behavioural changes with regard to public health measures, such as social distancing, wearing masks, self-isolation and so on, can be factored in the mathematical modelling predictions. Particularly, those infection-prevention measures are crucial in containing the virus when there have not been proper vaccines or therapeutics developed yet. According to Kim’s study, Behavioural-change (S_F_) and Hospital-quarantine (H) compartments are combined with the SEIR model [52]. The Behavioural-change compartment refers to a group of people who strives to avoid infection by those infection prevention intervention [53]. The model is comprised of Susceptible(S), Exposed(E), Behaviour-changed susceptible (S_F_), Exposed (E), Infectious (I), Hospital-quarantined(H), and Recovered (R) as illustrated in Figure 9.

The increased number of cases and strengthening of public health measures drives the Susceptible (S) group of people to move to the Behavioural-changed susceptible (S_F_) group. This could result in decreasing the probability of transmission in the Behavioural-changed susceptible (S_F_) group. The study also deals with the outbreak of Daegu and North Gyeongsang province (Gyeongbuk) where there was a dramatic surge in the number of cases at the same period. The mathematical modelling predicts that approximately 13,800 cases would occur across the whole country and the last case would be confirmed on June 14th, 2020. Particularly in the Daegu and Gyeongbuk regions, the cases were expected to reach approximately 11,400 and would end on 27 May 2020 (Figure 10) [52].

Both mathematical modelling predictions illustrate different results and patterns in association with pre-conditions and dynamics. Even though the government raised the highest level of infectious disease alert in the country and strict intervention policies have been implemented, it is necessary to find out whether those measures have been effective and maintainable as time has passed. Mathematical modelling is used as a useful tool in the decision making process in public health. Both mathematical modelling predictions commonly indicates the outbreaks in South Korea would continue at least until May, so the public health measures should be maintained.

## 4. Discussion

The WHO’s announcement of the coronavirus outbreak as the “pandemic” on 11 March 2020, followed the rapid spread of the virus around the world [54]. People were perplexed by the unprecedented propagation of a new viral disease and its huge impacts on public health as well as on multiple aspects of the global community. However, each country has been facing different circumstances as the responses to COVID-19 are immensely different among countries [55]. The findings in this case study, have shown how South Korea has responded to the COVID-19 with social, economic, political, and epidemiological impacts and it appeared to have impressive measures and results. As many other countries have been being criticized for poor outcomes regarding COVID-19, the Korean government had faced condemnation of the management of the MERS outbreak in 2015 that caused 186 cases and 38 deaths. Such experiences helped the government to establish improved strategies and measures against COVID-19. Consequently, it showed relatively better preparedness and outcomes than the rest of the world. South Korea encountered the epidemic earlier than other countries because of its geographic adjacency and international relationship with China. The first case was reported on 20 January 2020 and it did not take long to encounter community transmission which led to the peak of confirmed daily cases at 909 on 29 February. The new cases were mainly concentrated in Daegu and its surrounding Province, North Gyeongsang Province. The subsequent epidemiological investigation uncovered this large outbreak was associated with meetings of a fringe religious sect called Shincheonji. The Sincheonji-related outbreak accounted for nearly half (48.7%) of the total cases. As a result, South Korea was one of the most-affected countries with a high number of cases and deaths during the initial phase of the pandemic. However, South Korea has successfully flattened the curve of new cases after reaching the peak and the downward trend was maintained continuously. As of 30 April, the number of new cases per day were 4 and 84.2% of the confirmed cases were fully recovered. The number of confirmed cases and deaths were 10,765 and 247 respectively.

The Korean government has developed the plan and strategy to confront the outbreak of the new virus prior to its arrival. The government took the drastic and proactive intervention strategies including performing early and massive coronavirus tests, tracing contracts, and isolating cases instead of blocking the door completely against affected countries or putting the affected areas or whole population on lockdown. These strategies could be possible thanks to remarkable actions and innovation. For instance, since the government started the development of diagnostic test kits in collaboration with the private sector before having confirmed the first case in the country, it was able to produce test kits domestically at the early stage of the epidemic. Moreover, innovative ideas such as ‘drive-thru’ screening centres, the mobile Self-quarantine Safety Protection Application, the Living Treatment Centres were introduced. The drive thru screening centres gave an easy and safe way for both medical personnel and the public, thereby diagnostic tests could be conducted rapidly on a large scale. As of 30 April, the number of COVID-19 tests performed was 619,881. More importantly, Figure 3 shows the importance of early reactive case detection by comparison with the trend in tests conducted in the United States which reported the first COVID-19 case the day after South Korea confirmed its first.

As there has not been a single effective measure nor vaccine, the role of the public is also significantly important. Such importance of individual behaviours was taken into account the mathematical modelling. It cleared showed it significance in conjunction with tightened public health measures. Likewise, the Korean government emphasized the importance of the role of individual citizens and urged the public to abide by public health advices through effective risk communication. The government has provided the pubic with transparent information on the pandemic and the consistent message regarding wearing a mask and its effectiveness even when the World Health Organization stated that there was no sufficient evidence of effectiveness of masks and concluded not to recommend the use of masks against COVID-19 for the general population [56]. Moreover, the government actively intervened in the market to distribute enough masks to the people when there was a shortage of masks all over the country. Besides, South Korea improved affordability and accessibility of tests and treatment and undertook measures to prevent further community transmission such as continuous monitoring of the public and hospitals and designating hospitals where patients with a respiratory symptom were treated separately from the others (Figure 5).

However, South Korea has confronted the problems of the healthcare system during the pandemic. It has the poor public health sector, heavily relying on private providers and there is insufficient health workforce in general. Therefore, the government has mobilized army doctors and nurses, voluntary medical personnel, and public health doctors to mitigate the shortage of medical personnel. Especially, the public health doctors who work in a remote area instead of military service, played a significant role when the large outbreak occurred in Daegu and North Gyeongbuk Province.

The COVID-19 pandemic and measures have brought enormous impacts on the country. Given its economic partnership with China, the economic disruption was already observed before it spread nationwide in South Korea due to the slowdown of exports and weakened domestic demand. It also caused social conflicts and political debates especially when the Shincheonji-related large outbreak occurred. However, as the government has flattened the curve of numbers of new cases rapidly, it has regained the trust of the people which affected the result of the general election in April 2020.

## 5. Conclusions

Although several characteristics of the virus were revealed in the early epidemic case studies in China, the information of the virus was not fully discovered or not open clearly to the global society at the initial phase of the COVID-19 pandemic. Therefore, responses of each country to COVID-19 were highly divergent due to lack of coordinated guidelines worldwide and it has resulted in different outcomes among countries. South Korea is one of the countries that have shown better outcomes in terms of COVID-19 than the rest of the world. The interventions and strategies undertaken by South Korea appeared to be effective. The most distinctive feature is the drastic and proactive strategy. Instead of implementing traditional measures of containment of infectious diseases, the Korean government put emphasis on proactive case finding, contact tracing, and rapid isolation of cases. Moreover, it was underpinned by remarkable measures such as risk communication, civil society mobilization, and innovation. The second feature is that the Korean government did not implement travel restrictions except for arrivals from Wuhan, China, respecting the IHR of WHO while the rest of the world closed their borders rapidly [57]. Given the situations of the outbreak in the countries where travel bans or border closures were applied promptly, the question on the effectiveness of travel restrictions is still raised.

The key findings highlight the importance of the proactive strategy and the responses of South Korea to COVID-19 provide broadened insights. However, a further research is needed in order to understand the association between each measure and the outcome and the extent of its effectiveness before applying them in other countries. Therefore, it is also important to develop methods to measure and quantify the effectiveness of these responses.

As the pandemic is still ongoing, there is a compelling need to accumulate the scientific evidence and evaluate the full extent of performances of South Korea on the COVID-19 pandemic and social, economic, and political impacts after the pandemic.

## Figures and Tables

**Figure 1 ijerph-17-09571-f001:**
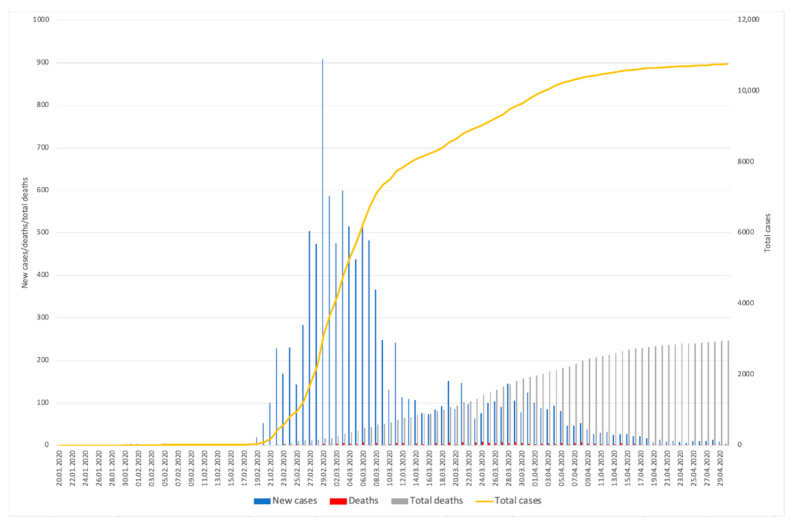
The trend in the number of confirmed cases of COVID-19 and deaths in South Korea. Based on data from Korea Ministry of Health and Welfare and the Statistic Korea under the Ministry of Strategy and Finance [26,27].

**Figure 2 ijerph-17-09571-f002:**
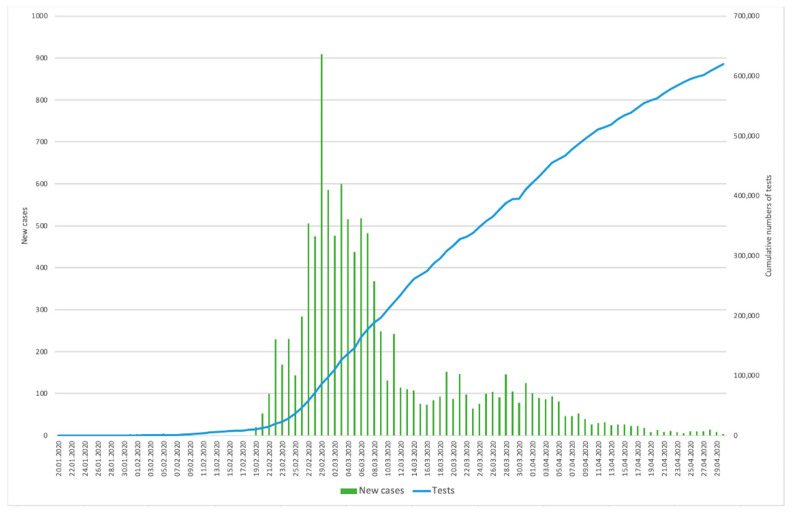
The trend in the number of confirmed cases of COVID-19 and cumulative tests conducted. Based on data from Korean Ministry of Health and Welfare and the Statistic Korea under the Ministry of Strategy and Finance [26,27].

**Figure 3 ijerph-17-09571-f003:**
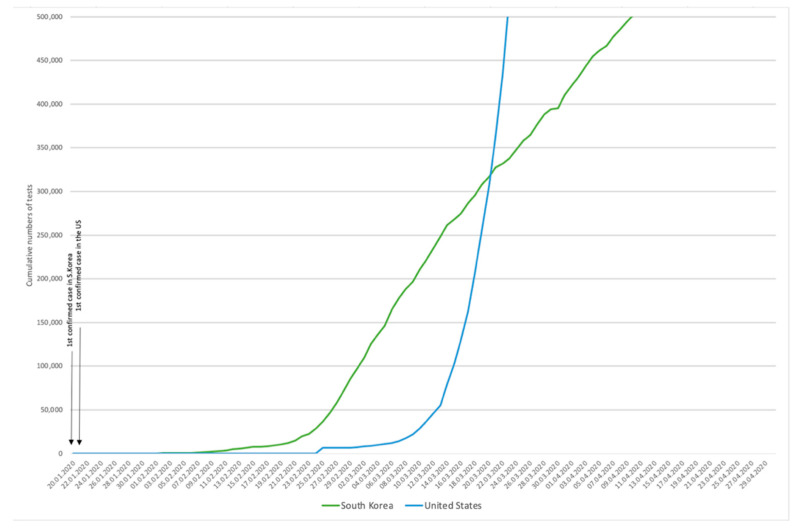
Cumulative numbers of test conducted in South Korea compared to the United States since the first case was confirmed both in their territories. Data retrieved from Korea Ministry of Health and Welfare, the US Centres for Disease Control and Prevention, and the COVID tracking project [26,29,30].

**Figure 4 ijerph-17-09571-f004:**
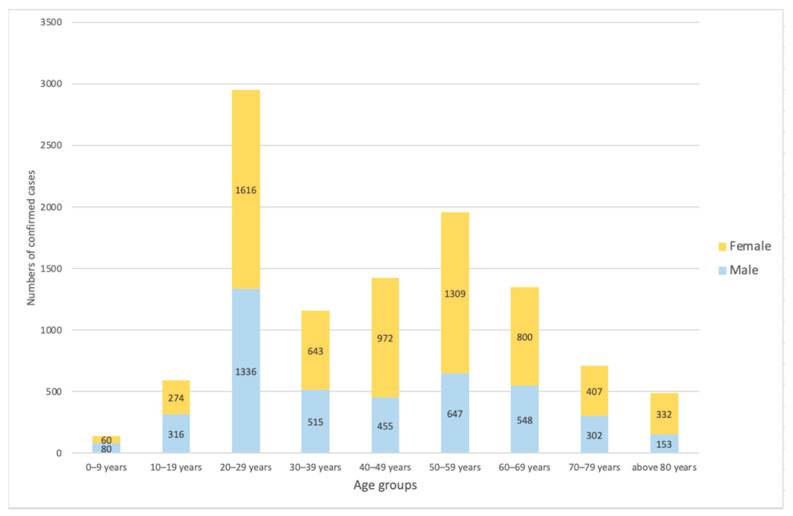
The number of confirmed COVID-19 cases by sex and age groups as of 30 April. Data retrieved from Korea Centres for Disease Control and Prevention [25].

**Figure 5 ijerph-17-09571-f005:**
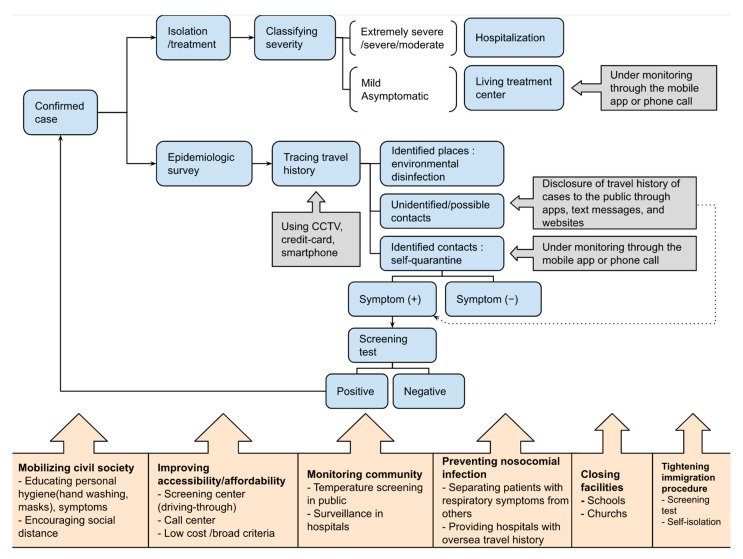
The schematic diagram illustrating interventions South Korea undertook to tackle COVID-19. It was developed to illustrate the non-pharmaceutical interventions of South Korea based on COVID-19 Patient treatment & management and quarantine system provided by Korea Ministry of Health and Welfare [34,36].

**Figure 6 ijerph-17-09571-f006:**
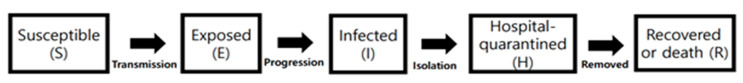
Flow diagram of SEIHR model for the COVID-19. Reproduced from [50], copyright 2020, the Korean Society of Epidemiology.

**Figure 7 ijerph-17-09571-f007:**
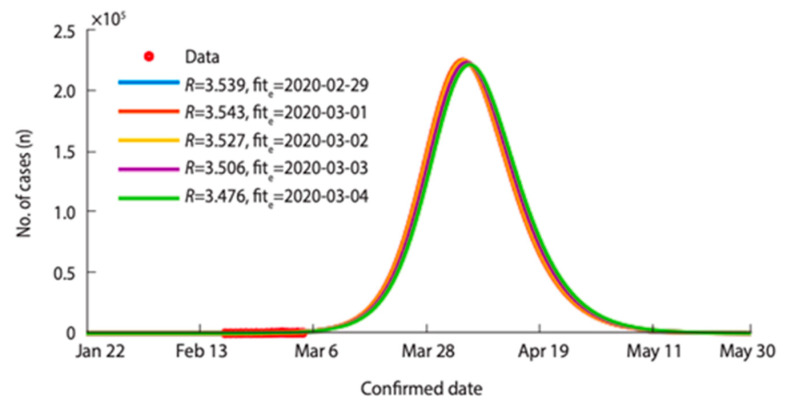
Estimated reproductive number by date (29 February–4 March 2020) and their estimated numbers of cases if there were no containment measure in Daegu and North Gyeongsang province [50]. Reproduced from [50], copyright 2020, the Korean Society of Epidemiology.

**Figure 8 ijerph-17-09571-f008:**
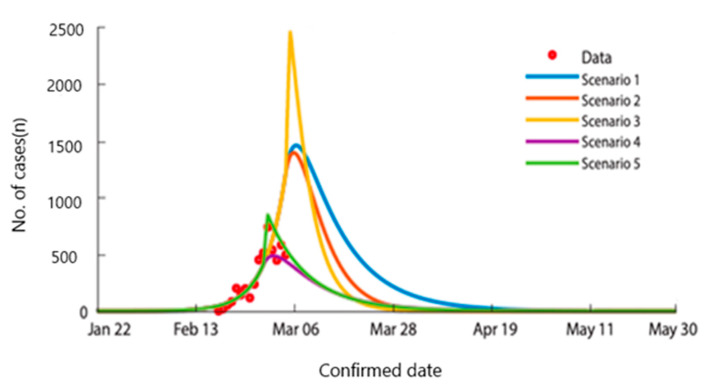
The predicted number of confirmed cases from February to March if there were proper containment measures according to mathematical modelling in Daegu and North Gyeongsang province [50]. Reproduced from [50], copyright 2020, the Korean Society of Epidemiology.

**Figure 9 ijerph-17-09571-f009:**
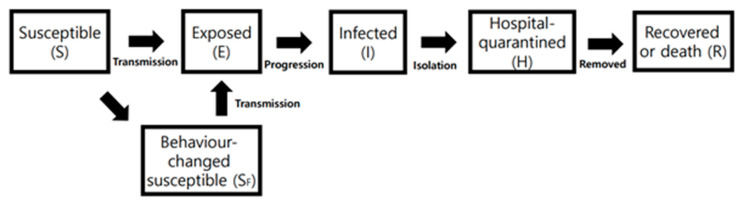
Flow diagram of the behavioural change added SEIHR model for the COVID-19 [52]. Reproduced from [52], copyright 2020, the Korean Society of Epidemiology.

**Figure 10 ijerph-17-09571-f010:**
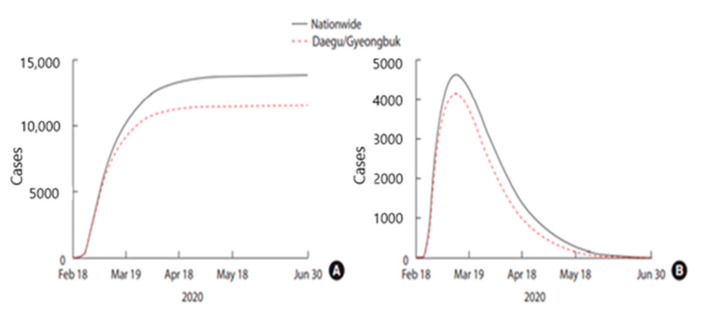
(**A**) Predicted cumulative confirmed cases over time, (**B**) Predicted isolated cases over time [52]. Reproduced from [52], copyright 2020, the Korean Society of Epidemiology.

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
