# Peer review of "Understanding South Korea’s Response to the COVID-19 Outbreak: A Real-Time Analysis"

_ijerph, 2020, doi:10.3390/ijerph17249571_

Round 1

Reviewer 1 Report

Ok.

Additional comments:

1. The theme of the article is important and current.But we must be very fine when it comes to filtering the information that is published.

 2. I consider it very important that the article needs a bibliographic justification with first-rate articles.

3. The conclusions must be consistent with the objectives set.

Reviewer 2 Report

I believe that the topic of the manuscript is interesting. I would like to suggest major revisions to this manuscript.

Specific comments:

1. Title

The title reflects the content and problem studied.

2. Abstract

The purpose of the study and the conclusions of the study were not presented in the abstract. Please add them in the abstract.

3. Key Words

The keywords are representative of the subject studied and exposed. 

4. Introduction

This is not a government report, so the background for research should be presented academically.

5. Core part

Please present the research procedure and rationale in more detail. In particular the rationale for the COVID-19 response strategy presented in Figure 3 should be suggested with reasonable references.

6. Conclusions and Discussion

The discussion section should not be a suggestion of new contents about COVID-19. In this section, there should be a discussion using bibliographical references to key findings in core part in your manuscript.

7. Suggesttion of improvements and implications

Based on the results of this study, it is necessary to suggest improvements and implications for government and practitioners, and future research projects.

8. References

Any missing reference year and DOI will need to be inserted.

Reviewer 3 Report

The manuscript is well studied and a few minor corrections below will help to improve the quality of the manuscript.

  1. The text is generally well written and comprehensive to the readers. however, each section has too many short paragraphs, I would recommend to merge some of those paragraphs.
  2. The authors did not used sufficient references. Add and include recent literature references.
  3. Concentrate the graphical representation of all figures in a pleasing way to attract the readers.
My additional comments:   Discuss in detail about diagnostic and treatment strategy which is used by south korea is more useful.

Reviewer 4 Report

See the attachment

Round 2

Reviewer 2 Report

The authors have faithfully corrected what the reviewer pointed out.

It's hard to see the numbers in the figures on pages 5-6. Adjust them.